# Is There a Deficit in Product and Process of Handwriting in Children with Attention Deficit Hyperactivity Disorder? A Systematic Review and Recommendations for Future Research

**DOI:** 10.3390/children11010031

**Published:** 2023-12-27

**Authors:** Frédéric Puyjarinet, Yves Chaix, Maëlle Biotteau

**Affiliations:** 1Montpellier Psychomotor Training Institute, UFR de Medicine Montpellier-Nîmes, University of Montpellier, 34090 Montpellier, France; 2Pediatric Neurology Unit, Children’s Hospital, Toulouse University Hospital Center, 31059 Toulouse, France; maelle.biotteau@inserm.fr; 3Toulouse NeuroImaging Center (ToNIC), University of Toulouse, INSERM, 31024 Toulouse, France

**Keywords:** ADHD, handwriting, dysgraphia, product of handwriting, process of handwriting

## Abstract

Handwriting abnormalities in children with attention deficit hyperactivity disorder (ADHD) have sometimes been reported both (i) at the product level (i.e., quality/legibility of the written trace and speed of writing) and (ii) at the process level (i.e., dynamic and kinematic features, such as on-paper and in-air durations, pen pressure and velocity peaks, etc.). Conversely, other works have failed to reveal any differences between ADHD and typically developing children. The question of the presence and nature of handwriting deficits in ADHD remains open and merits an in-depth examination. The aim of this systematic review was, therefore, to identify studies that have investigated the product and/or process of handwriting in children with ADHD compared to typically developing individuals. This review was conducted and reported in accordance with the PRISMA statement. A literature search was carried out using three electronic databases. The methodological quality of the studies was systematically assessed using the Critical Appraisal Skills Program (CASP) criteria. Twenty-one articles were identified. Of these, 17 described handwriting quality/legibility, 12 focused on speed and 14 analyzed the handwriting process. All the studies (100%) with satisfactory methodology procedures reported an impaired product (for quality/legibility) and 91.7% reported abnormalities in process, while only 25% evidenced a difference in the speed of production. Most importantly, the studies differed widely in their methodological approaches. Substantial gaps remain, particularly with regard to ascertaining comorbidities, ADHD subtypes and the medical status of the included children. The lack of overall homogeneity in the samples calls for higher quality studies. We conclude with recommendations for further studies.

## 1. Introduction

### 1.1. Attention Deficit Hyperactivity Disorder (ADHD)

Attention deficit hyperactivity disorder (ADHD) is a neurodevelopmental condition characterized by marked symptoms of inattention and/or impulsivity–hyperactivity [1,2] in children with preserved intellectual abilities in the absence of any physical or sensory abnormalities. ADHD affects around 5–7% of children [3,4,5] and involves developmentally extreme and cross-situational displays of (a) inattention and/or (b) hyperactivity–impulsivity that manifest in more than one setting (e.g., home, school, sport, leisure or other social environments). The DSM-5 criteria define four presentations of ADHD: inattentive (ADHD/I), hyperactive–impulsive (ADHD/HI) and combined presentations (ADHD/C). Other forms are classified as unspecified ADHD.

It is a lifelong disorder and around one child out of two will still experience symptoms in adolescence and adulthood [3,6]. The neurodevelopmental etiology is no longer debated [7,8], although many environmental risk factors are known to interact with a genetic susceptibility [2]. Comorbidities are common [9], with autism spectrum disorders (ASD), psycho-emotional disturbances and developmental coordination disorder (DCD) being the best-known examples. Children with ADHD often experience significant academic impairments [10], and 45% meet the criteria for a comorbid learning disability [11,12].

### 1.2. Handwriting Deficits in ADHD

Among the learning difficulties, researchers have been paying increasing attention in recent years to handwriting problems, which often include a lack of legibility in letter-form production, spacing, spelling and syntactic and composition disturbances, whether or not these are associated with insufficient speed production. These characteristics are generally encapsulated under the generic term “dysgraphia”. However, some authors have suggested a more precise definition for this disorder, which is mainly based on having impaired letter-form production through the hand and is, therefore, focused on quality/legibility (e.g., [13]). In line with this perspective, Hamstra-Bletz and Blöte [14] had already defined dysgraphia as a written language disorder that affects the mechanical writing skills of children with no distinct neurological deficit.

Currently, a dysgraphia diagnosis implies the handwriting product and process evaluation (Figure 1). The product refers to the static features of the written trace, such as letter form and size, spatial organization of the text, number of erasures, etc. A quantitative measure of handwriting speed is also considered, mainly based on the number of characters written in a given period of time. The process of handwriting describes the analysis of the dynamic and kinematic components involved in the movement of writing. A number of variables can be analyzed: cognitive abilities (e.g., working memory, inhibition), posture, finger and arm movements, pen grip and finger pressure on the pen, in-air and on-paper durations, pen velocity, pen pressure, etc. Several handwriting processes can be assessed via digitizing tablets, as has been performed in a growing number of studies (e.g., [15,16,17]). 

Some studies have suggested that 50 to 70% of ADHD children demonstrate disturbances in their handwriting legibility and speed [21,22,23,24]. A greater variability; slowness of writing; poor rhythm and flow of writing; poor organization of the written material; poor alignment; poor overall legibility; pronounced variability in the spatial components; poor spacing within and between words; poorly formed letters; inconsistent letter size and shape; letter omissions, insertions, inversions or substitutions; and frequent omissions of words or frequent erasures have all been reported (see [23,25,26]). However, when examining these studies more in detail, it becomes apparent that they provide unclear or even discordant results. One striking example concerns writing speed: the conclusions drawn from a comparison between ADHD and typically developing children are contradictory. Some works have demonstrated no difference [27], while others have found that children with ADHD write more slowly [28,29,30,31]. Other studies have even revealed that children with ADHD write faster [21,32]. How can such results be explained? Are the studies really comparable? Are there confounding variables that the authors did not consider? In addition, it seems difficult to extract the exact number of ADHD children who display handwriting impairments due to the apparent paucity of studies examining this aspect. Finally, the fact that several studies evaluated ADHD children who were on medication while others did not adds confusion to the overall picture, because handwriting skills may be sensitive to methylphenidate [21,33]. 

### 1.3. Aim of the Systematic Review

Taken together, there is partial evidence for handwriting abnormalities in subjects with ADHD, although the results are often equivocal, maintaining a certain vagueness. A systematic review is, therefore, needed to examine the quality of the evidence as well as include the relevant studies up to 2023 that used paper-and-pen assessments, questionnaires and/or digitizing tablets. To sum up, the specific objectives of the present work are to (i) conduct a systematic review of the ADHD literature focusing on handwriting skills; (ii) examine the methodological quality of the relevant studies; (iii) describe whether the evidence for a deficit in the handwriting product and process is convincing enough to conclude that children with ADHD have dysgraphia; (iv) determine whether all children with ADHD are affected; and (v) make informed recommendations for future research.

## 2. Method

### 2.1. Search Strategy

To include all the relevant articles in this systematic review, a search was carried out using the PubMed, Web of Science and CENTRAL electronic databases, with no restrictions on the year of publication and only limited to English-language articles. We selected these databases for their broad spectrum of disciplines, which regularly publish research pertinent to the topic of this review for ADHD. Manual searches were also conducted to find further references to appropriate articles. The final search included publications dating up to September 2023. The following keywords were inputted: (“handwriting” OR “dysgraphia” OR “written production” OR “fine motor abilities” OR “fine motor skills”) AND (“attention deficit hyperactivity disorder” OR “ADHD”) AND (“children” NOT “adults”).

### 2.2. Identification

The database search pinpointed a total of 814 records. After removing duplicates (*n* = 61), a total of 753 records were identified. On the basis of the abstracts, titles and in- and exclusion criteria, 36 potentially relevant articles were recognized. Based on the full text, 16 of these 36 were selected for this systematic review, and were supplemented with five articles found in the reference lists. This resulted in a total of 21 included articles. Twenty were case–control studies while one was a retrospective cohort-based study. Details can be found in the flow chart of the included and excluded studies (Figure 2).

### 2.3. Study Selection

An evaluation of the titles and abstracts was conducted to decide whether or not the articles were eligible for this review. The inclusion criteria were that the publications had to (1) report data linked to handwriting characteristics in children with ADHD regarding the product and/or process (e.g., legibility, spatial components, correction errors, letters size, speed of handwriting, amplitude of movement, in-air time and other kinematic features, pen pressure, etc.); (2) contain data on handwriting characteristics whether or not the children had taken methylphenidate and regardless of the presentation of ADHD (e.g., inattentive or hyperactive–impulsive presentation); and (3) provide a comparison between children with a formal diagnosis of ADHD according to international criteria (e.g., based on DSM-5; [1]) and a typically developing control group. The exclusion criteria were (1) qualitative and case studies; (2) no handwriting measures; (3) the absence of a typically developing control group; (4) the absence of a formal diagnosis of ADHD; and (5) subjects older than 18 years of age.

### 2.4. Methodological Quality

All included publications were evaluated using the Critical Appraisal Skills Program (CASP) dedicated to experimental studies [34]. The CASP questionnaire enables the assessment of a study’s validity via three main sections by asking the following questions: (1) Are the results of the study valid? (Section A); (2) What are the results? (Section B); and (3) Would the results help locally? (Section C). In this way, the methodological quality, presentation of results and external validity are systematically examined in order to check whether comparisons may reasonably be made between one study and another if necessary. A few adaptations have been proposed in terms of the formulation for acquiring a rapid answer (Yes, No or Cannot Tell) to the questions which are listed in Table 1. The results of the validity between studies are displayed in Table 2.

## 3. Results

The characteristics of the included studies are presented in Table 3, with the first author, year of publication, sample size, mean age, gender, inclusion and exclusion criteria, ADHD presentation, medication state, and handwriting measures. The statistically significant main results are reported in Table 4. Figure 3 provides a quick summary as to whether the product and/or process are impaired in children with ADHD when compared to typically developing subjects.

For each domain (product and process of writing), the study characteristics, methodological quality and results are discussed. Questions 6 (*Have the authors taken account of the potential confounding factors in the design and/or in their analysis?*) and 9 (*Do you believe the results?*) on the CASP checklist were essential for deciding whether studies should be retained before drawing conclusions. These methodological considerations prompted us to analyze the conclusions of each study twice (see Flow Chart, Figure 2), before (Step 1) and after (Step 2) excluding those which were not sufficiently satisfactory for each domain studied. Figure 3 states the high methodological biases per domain for each study by means of a warning symbol. Figure 4 and Figure 5 show the results for each domain using pie charts for both the Step 1 and Step 2 analyses.

A great heterogeneity is observed in terms of the sample characteristics, assessment tools and medication status. The inclusion and exclusion criteria for the ADHD groups varied across the studies. Some authors only excluded potential comorbid reading problems (e.g., [28]), whereas others were much more restrictive and excluded intellectual disabilities; methylphenidate medication; learning disabilities; mathematical or reading disorders; and neurological, sensory, motor, psychiatric or mood disorders (e.g., [36]). ADHD presentation was either specified (e.g., [27,30]) or not stated (e.g., [46,48]). Some authors mentioned whether methylphenidate was taken (e.g., [42]) while others did not (e.g., [40,47]) and, in one study [49], the handwriting skills of the ADHD children were tested twice, with and without methylphenidate. It is important to observe the wide diversity of assessment tools and conditions: paper-and-pen materials (e.g., [38], digitizing tablets (e.g., [45]) and even questionnaires for the parents [35] were proposed to assess handwriting characteristics. With respect to writing conditions, spontaneous letter production [45], copy tasks (e.g., [32]) or dictation tasks (e.g., [37]) were suggested.

### 3.1. Product of Handwriting Results

#### 3.1.1. Quality/Legibility

Seventeen out of twenty-one studies examined quality/legibility [27,28,30,32,35,36,37,38,39,40,41,42,43,46,47,48,49,51]. The findings of six studies could not be considered for Step 2 as a result of major methodological biases: the study by Farhangnia et al. [40], for the absence of inclusion/exclusion criteria; the study conducted by Flapper et al. [41], due to the associated DCD for all the ADHD children, making it impossible to determine whether handwriting difficulties resulted from ADHD per se or the DCD; the study by Frings et al. [42], owing to the absence of clear exclusion criteria; the studies by Laniel et al. [46], Okuda et al. [48], and Rosenblum et al. [32], on account of the insufficiently detailed inclusion/exclusion criteria and reduced sample size (*n* < 15), thereby implying the presence of potential critical confounding factors and methodological weakness. Tucha and Lange [49] pointed out two results: the ADHD children showed a significantly poorer quality of handwriting without treatment than the boys in the control, but presented no difference with methylphenidate. All the studies (100%) reported differences between the ADHD children and the control groups either before (17/17 studies) or after (11/11) exclusion.

#### 3.1.2. Primary Conclusions with Respect to Quality/Legibility of Handwriting

Beyond the observed methodological heterogeneity and after the exclusion of works with important biases, it is reasonable to assume that handwriting quality is indeed impaired in ADHD. Nevertheless, very few studies have reported effect sizes, making it impossible to precisely quantify the significance of these difficulties. Finally, it is not possible to tell whether all ADHD children manifest an impairment in the quality/legibility of their handwriting due to an absence of individualized data, which could uncover potential inter-individual variability in writing performance. 

#### 3.1.3. Speed of Handwriting

Twelve included studies out of twenty-one focused on the speed of handwriting [27,28,30,32,36,37,38,39,40,41,43,46,50]. After analyzing the findings independently of methodological quality, 8 out of 12 studies (66.7%) reported no difference between the ADHD children and the control groups, versus the 33.3% in favor of a variation in writing speed. The latter proportion dropped to 25% in Step 2 after studies with major methodological biases were excluded (i.e., [32,40,41,46]). The only work showing a significant difference, with a slower writing speed in the well-identified non-medicated ADHD children, is the study by Borella et al. [36]. In the research conducted by Hung and Chang [30], it was unclear whether or not the ADHD children were on medication, which hindered our ability to draw a clear conclusion. 

#### 3.1.4. Primary Conclusions with Respect to Speed of Handwriting

After excluding the studies with major biases, the trend, therefore, pointed towards an absence of difference in handwriting speed between unmedicated children with ADHD and typically developing subjects. As observed earlier, we cannot state whether all children with ADHD manifest problems in the speed domain, due to an absence of individual data in the included studies. The overall results considering the product (i.e., quality/legibility) and the speed of handwriting before and after the exclusion of studies with major methodological biases are displayed in Figure 4.

### 3.2. Process of Handwriting

Fourteen studies out of twenty-one examined the handwriting process [27,28,30,31,32,36,37,38,43,44,45,46,49]. There were various targeted variables: working memory load, strokes duration, ballisticity, in-air time and pen pressure. Only one study [43] reported the absence of difference (considering the coefficient of variability in phrase height and width). Tucha and Lange [49] found that methylphenidate use led to the handwriting process’s deterioration, but following withdrawal, the results of the ADHD children did not differ from that of the control groups. Before exclusion (Step 1), 13 out of 14 studies (92.9%) indicated variations between the ADHD children and the control groups. This score remained at 91.7% after Step 2, with the exclusion of studies by Laniel et al. [46] and Rosenblum et al. [32], for the same reasons as mentioned previously (Figure 3). Authors have highlighted that ADHD children demonstrated increased pen pressure [28], greater variability in acceleration–deceleration phases [36] and in stroke length [44], or more inversions in the direction of their velocity profiles, thereby indicating a lack of automation [31] when compared to the control groups. 

#### Primary Conclusions Regarding the Process of Handwriting

The evidence clearly favors an impaired handwriting process in children with ADHD. When available, the effect sizes indicated a significant impact of ADHD on the handwriting process, thereby highlighting important disturbances beyond the product per se (e.g., [27,30,37,38]). In regard to the evaluation of the handwriting product performance, the studies did not provide any individual data that would have enabled us to confirm any inter-individual variability in the handwriting process. The results considering the handwriting process before and after the exclusion of studies with major methodological biases are displayed in Figure 5.

## 4. Discussion

The two main objectives of this systematic review were to (i) describe whether the evidence for a deficit in the writing product and process is compelling enough to conclude that children with ADHD manifest dysgraphia, and (ii) determine whether all children with ADHD are affected.

### 4.1. Is Handwriting Performance in Children with ADHD Really Impaired?

This systematic review shows that 100% (17/17 for Step 1, 11/11 for Step 2) of the studies comparing ADHD and typically developing children reported an altered quality of the written trace in ADHD individuals, 33.3% (4/12 for Step 1) and 25% (2/8 of studies for Step 2) revealed an altered speed of production, while 92.9% (13/14 for Step 1) and 91.7% (11/12 for Step 2) of the studies described an impaired handwriting process. 

ADHD children therefore clearly seem to experience problems both with the product (mainly for quality) and process of handwriting. These results, which show that both domains come under impairment, are in line with recent works on typically developing school-aged children supporting the idea that handwriting quality and speed significantly correlate with various process characteristics (e.g., the number of strokes, reaction time, duration, on-paper duration, pen pressure, vertical and horizontal sizes, absolute velocity, etc.). In the study by Coradinho et al. [52], poorer handwriting quality was notably associated with a higher average absolute pen velocity, larger vertical or horizontal sizes and lower relative on-paper duration. This suggests that kinematic abnormalities could at least partly account for difficulties in terms of the quality and/or speed of handwriting. If we consider writing performance along a continuum, such associations between handwriting quality and kinematic variables may be even more pronounced in ADHD children. In our review, the finding that around 100% of studies detected abnormalities in the handwriting product and process of children with ADHD compared to the control groups also suggests close links between the two spheres. It is important to note that effect sizes–when available–indicated a considerable impact on process due to ADHD (e.g., [27,30,37,38]). However, all these observations do not really stand up to scrutiny when considering writing speed. Indeed, only 25% of the studies with a satisfactory methodology (Step 2) reported a difference in handwriting speed between ADHD and typically developing children. This calls for caution and further studies with better methodological quality for clarifying the characteristics of ADHD subjects in the domain of handwriting speed.

Our results, overall, should be considered with great caution. Firstly, very few studies reported effect sizes when considering the product. It is, therefore, extremely difficult–if not impossible–to determine whether the differences observed between ADHD children and the control groups are important or not. Moreover, handwriting problems associated with ADHD might be due to a comorbid DCD where handwriting difficulties are well identified [12,53,54,55,56]. More generally, comorbidities have not been screened for rigorously in studies, although their impact on the handwriting skills of ADHD children may be crucial. In addition, since most studies did not use standardized tools, it is also difficult to know whether ADHD children display mild difficulties or severe dysgraphia. The approach of identifying handwriting difficulties along a continuum ranging from mild-to-severe dysgraphia is gaining support. From this perspective, recent studies have argued against a dichotomic classification of children as non-dysgraphic on the one hand or dysgraphic on the other (e.g., [53]). Additional works will have to detect where each ADHD child is situated along this continuum. Finally, although there is no gold standard for diagnosing dysgraphia, it has to be noted that a number of qualitative tests have been developed [57] which assess both the product and process with available norms and acceptable reliability [58]. Our review shows that some more subjective or esoteric evaluations were used instead, hindering the comparability of the results.

An intriguing question, even if it is out of the scope of our review, lies in the putative beneficial effects of methylphenidate on the handwriting skills of children with ADHD. At best, medication seems effective for a portion of children (e.g., [22]) while, at worst, there is no impact on quality but rather on speed, which is often slowed down, and a change in process for some children (e.g., [49,59]). Again, such equivocal results highlight inter-individual variability regarding the mechanisms which underpin handwriting disturbances. In some people with attentional and executive deficits, which are very common in ADHD, handwriting problems could be the direct consequence of impoverished cognitive control. In this case, methylphenidate could largely contribute to improving handwriting skills, although fluency seems to deteriorate in parallel (see [49]). Overall, such contradictory findings suggest that there is a need to identify ADHD children who take (or do not take) methylphenidate or other drugs in studies investigating handwriting skills, given the possible beneficial effect for a number of subjects. From a clinical point of view, it is also very important to realize that methylphenidate will not automatically improve handwriting quality, and may even contribute to slowing down the speed of production. This warrants an individualized approach for each child when considering all the parameters involved in handwriting, notably cognitive functioning, the degree of severity of the handwriting difficulties, potential methylphenidate consumption, alteration in product and/or process, etc.

In summary, and in response to the question “*Is handwriting performance in children with ADHD really impaired?*”, we can, therefore, answer that yes, in light of this review, there do seem to be difficulties linked to the written trace in ADHD children, especially for quality/legibility. However, almost nothing is known about the degree of severity of these difficulties. Moreover, there is an evident paucity of data regarding the proportion of children with ADHD experiencing impairment in written trace production. Finally, the tendency is to admit that there is no obvious difference in handwriting speed between ADHD and typically developing children, but further studies are essential in this area to refine the results.

### 4.2. Are All Children with ADHD Affected by Handwriting Deficits?

From our review, it is evident that children with ADHD encounter more handwriting problems than non-ADHD children. However, we cannot know the proportion of ADHD children affected by handwriting difficulties, since almost all the studies failed to consider potential inter-individual differences. The exception was the study by Lofty et al. [47], which reported that 50% of ADHD children in their sample experienced mild-to-moderate difficulties. We are faced with a major issue here, since a plethora of studies have showed that significant inter-individual variability of outcomes and performance in diverse tasks and contexts is a hallmark of ADHD [60]. It is, therefore, highly probable that all children with ADHD do not present the same level of written performance, although this remains to be demonstrated beyond the study conducted by Lofty et al. [47]. This lack of data is particularly regrettable, given that for other neurodevelopmental conditions the picture is clearer and helps with an overall understanding of children’s difficulties. In the case of DCD, for example, up to 87–88% of children have handwriting disorders, with around 15% experiencing a severe deficit (e.g., [53,54,61]). Generating the same type of evidence for handwriting skills in ADHD is, therefore, fundamental to support medical care decision making and the support required at school. Yet, these difficulties in identifying the prevalence of ADHD children affected by mild handwriting difficulties or severe dysgraphia fall within a more general framework. In truth, it is obvious that the lack of a clear and consensual definition of dysgraphia hinders a reliable estimation of its worldwide prevalence. Estimates of school-age children with dysgraphia range from 10 to 30% [62,63,64] depending on the definitions used. The disorder is marked by a dearth of precise criteria sets for diagnosis and, according to DSM-5, dysgraphia can be diagnosed as an “impairment in written expression” [1], which is a very broad definition, leaving plenty of scope for subjective views. In studies on writing impairments, different definitions of dysgraphia are, therefore, used, but only 5% of children would be included if limited to strict handwriting difficulties [65]. A recent study even found that only 41% of children affected by handwriting difficulties are impaired enough to use the term dysgraphia [66], thereby drastically reducing the prevalence of the disorder. It seems duly urgent to clarify the criteria characterizing handwriting difficulties that could culminate in dysgraphia if severe and persistent enough.

In response to the question “*Are all children with ADHD affected by dysgraphia?*”, we can, therefore, answer that the estimated proportion is still unknown, given the evident paucity of data which came to light through our review.

### 4.3. Suggested Recommendations for the Conduct of Studies on ADHD and Comorbid Handwriting Deficits

The broad range of handwriting impairments across all the included studies could reflect discrepancies in letter forms combined with various handwriting educational background systems in different countries [67], but it is likely to mirror variations between the experimental methods used. There are indeed a number of studies where the methodological approaches were deemed to introduce possible biases into the results. Overall, a key finding of our review is that standardized procedures for the conduct of studies in this field are needed. To our knowledge, there are no known guidelines for carrying out studies in dysgraphia comorbidity in general or in strict co-occurrence with ADHD. On the basis of observations arising from our review, and completed using the Clinical Practice Guideline for the Diagnosis, Evaluation and Treatment of ADHD [5], we have, therefore, provided some recommendations for future studies in this domain.

#### 4.3.1. Dysgraphia Evaluation

While ADHD diagnosis criteria were sufficient overall in almost all the studies, the parameters for dysgraphia case inclusion were not clear. Firstly, according to the studies, “dysgraphia” terminology may be used to encompass several disorders, ranging from strict handwriting to spelling or reading. Secondly, the profile of the children included varied greatly, depending on the selection criteria and assessment instruments, while the severity of ADHD was not considered. Thirdly, in a number of the included studies, handwriting performance was evaluated using informal qualitative observations performed parents and/or teachers. There are, as of now, a variety of objective measures (formal quantitative standardized tests) with which to judge children’s handwriting performance at different ages, and which measure both the legibility and speed of handwriting with acceptable reliability [58]. Although observations from both parents and teachers are helpful, self-rated questionnaires can be insufficient (sometimes asking parents to answer only one general question), imprecise (most parents do not possess adequate knowledge for comparison purposes) and, above all, too subjective. We recommend the use of standardized, valid and reliable tools that provide a quantitative score to determine if children are affected by a handwriting disorder outside the normal range and the severity of impairment. 

The issue of those in charge of measuring children’s performance also requires consideration. Even when excluding teachers or parents for the abovementioned reasons, only one evaluator, sometimes with unreported areas of competence, was probably found to assess handwriting skills. This measurement bias could be prevented two-fold, by using the expertise of a handwriting specialist and employing a double-blinded method. Accordingly, the examiner should not be informed about whether or not the children have comorbid ADHD and handwriting disorders. Given the subjective nature of some criteria, the use of two independent judges also seems requisite, ideally providing additional intra-class correlations for ensuring homogeneity in the scoring procedure. It should also be noted that a coupled product and process analysis is possible when the writing is recorded on digitizing tablets. Several kinematic variables can then be computed (e.g., pen grip and finger pressure on the pen, in-air and on-paper durations, velocity, etc.) more objectively [68,69]. We recommend a minimum of two independent, blinded, trained judges, with expertise in handwriting assessment, to assess the handwriting process and product in a less subjective manner. The use of digitizing graphic tablets should be favored. New tools, such as deep learning procedures for detecting dysgraphia, are also under development [70] and should improve the scoring procedure in years to come.

Finally, the experimental tasks given to the children varied hugely: writing a continuous repetitive alternated sequence of cursive letters, numbers, words, sentences or text; writing on lined paper sheets, on blank pages or digitizing tablets; and production/composition, dictation, and copying (near-point or far-point copying) tasks, under working memory or cognitive load, etc. This broad variability compromises the comparability of findings. It is of crucial importance to harmonize the measurement of key handwriting elements, and to use common outcome measures to facilitate the pooling and comparison of study findings. In addition, experimental methodologies could sometimes fail to represent real school life experience. Studies need a non-artificial evaluation which captures a child’s performance in everyday life settings (i.e., in the most environmentally friendly conditions possible). We recommend that experimental tasks be as similar as possible from one study to another, and that they represent the child’s experiences at school or at home as closely as possible in order to highlight his or her real writing difficulties. The use of longitudinal studies could also provide valuable information, as they enable the collection of very detailed information without intervention. Handwriting data could be gathered as part of routine care procedures in standard medical practice instead of in experimental frameworks. It seems primordial, of course (to ensure the comparability of results), to use matched comparison groups which require, for these types of comorbid studies (ADHD + handwriting deficit), a control group with typically developing children, another containing ADHD children only and a last group comprising children affected only by handwriting disorders.

#### 4.3.2. Confounding Factors

One of the most striking results of our review was that few studies observed the same handwriting impairments. While the administration of different tasks contributes to this heterogeneity, it does not account for the whole picture. An explanation may also lie in the heterogeneity of the samples. In reality, the vast majority of the studies failed to explore ADHD subtypes or comorbidities. ADHD often co-occurs with other neurodevelopmental disorders, psychiatric disorders (depression and anxiety disorders) or sleep disturbances [71]. If their presence does not rule out a diagnosis of ADHD, such comorbidities could have a real impact on handwriting skills and, therefore, may induce major biases. Handwriting deficits are actually not specific to children with ADHD, and may be recognized in other disorders often comorbid with ADHD, such as depression, sleep deprivation, or in DCD [72,73], among others. The clinical presentation of ADHD (inattentive, hyperactive–impulsive or combined) may also play a role. Patterns of associated disorders differ between ADHD subtypes, with ADHD inattentive being more strongly associated with academic impairment and manual dexterity deficits, while hyperactive–impulsive symptoms are mainly linked to eagerness and rushing [74]. Handwriting abnormalities are also linked to the severity of ADHD and, the more problematic the symptoms, the poorer the handwriting performance [75]. Studies that have identified more subtypes than those present in the DSM-5, with different levels of ADHD symptom severity, support this view. For example, Elia et al. [76] used latent class analyses procedure and highlighted six clusters among 500 individuals: one with ADHD children manifesting severe combined symptoms, two clusters with moderate symptoms, one with mild combined symptoms, one with moderate inattentive signs and mild hyperactivity and, finally, one with severe inattentive symptoms and moderate hyperactivity. Ideally, the knowledge that there may be more subtypes than those described in the DSM-5, with various symptoms severity, should prompt investigators to precisely characterize the profile of each ADHD child included in future studies, as subsequent handwriting skills may depend directly on these profiles. In this sense, we could speculate, for example, that children with profiles integrating severe inattention symptoms might be slower in terms of production speed, while children with combined profiles but of mild severity might write faster. In the same vein, regarding the handwriting process, we might expect that ADHD children with severe attention deficits—and, therefore, with probable associated severe manual dexterity impairment—might show more problematic qualities than children with other symptom profiles of different severity. Clearly, these possibilities have not been sufficiently taken into account in the studies included in our review.

The choices of age ranges as well as gender distribution were also insufficiently explained in different studies, for handwriting ability acquisition is a long process [19,20]. In addition, gender is known to have an impact at least on the handwriting product (legibility) in typically developing children [77]. Socioeconomic factors can also alter handwriting skills [78], as well as ethnicity and cultural background [67]. Therefore, we recommend that individual and demographic factors associated with ADHD or handwriting skills are properly identified and considered in order to minimize possible biases: all possible comorbidities (neurodevelopmental, psychiatric), ADHD presentation, age, gender as well as ethnicity, cultural background, socioeconomic status and familial handwriting habits. Even if handedness has not been identified as a predictor of handwriting quality [79] or writing speed [78], its impact on the handwriting process has been sufficiently studied and this factor should take precedence in future studies.

#### 4.3.3. Medication and Behavioral Treatments

Among ADHD children, a substantial number take medication (methylphenidate continues to be the first-line medication) and/or benefit from behavioral treatment (diverse home-based and school-based behavioral treatments, psychosocial treatment, training interventions, psychoeducation, learning and academic support, parental practices, school accommodation, intervention for management of associated symptoms, etc.). Most worldwide medical organizations suggest beginning with psychoeducation and behavioral management and, thereafter, the use (additionally or not) of psychostimulant medications [80]. Only US guidelines recommend medication as the initial treatment and, consequently, 60 to 70% of school-aged American ADHD children are taking medication [81]. If more data are needed to judge the efficacy of all the existing non-medication treatments, a large number of meta-analysis studies found medications to be highly effective in reducing ADHD symptoms or associated impairments, including in the handwriting domain [8,82]. The influence of medication on motor skills (dynamic balance and fine motor skills) is particularly well demonstrated [25]. However, results are more divergent concerning handwriting according to the authors. As a result of our review, we share the opinion that more evidence is needed to affirm that medication has a positive influence on handwriting, though [82] found in their systematic review that medication is effective for ADHD children who manifest comorbid reading disorders. Too few studies have considered medication in their analysis and conclusions, while none have appraised the possible confounding effects of behavior management on handwriting. Consequently, contemplation of all treatments, past or present for both ADHD and/or comorbid symptoms, is strongly recommended to observe whether children with ADHD who are treated possess different handwriting features when compared to their matched peers who are not. Most significantly, the interaction between handwriting skills and medication should be addressed with great attentiveness, given that their effects on symptoms beyond the strict framework of ADHD are well documented. Once again, the use of real-life longitudinal studies would be a major asset, as they would make it possible to compare groups of treated subjects to untreated subjects, or make pre–post treatment observations in order to highlight the correlation between the treatment under consideration and the evolution of the handwriting disturbances. We recommend that future studies scrupulously identify and consider all past or present medications and non-medication treatments.

### 4.4. Theoretical Considerations

We close this section with the following aspects that seem important. It would be relevant to support clinical findings with more fundamental work dealing with the theoretical models of handwriting, whether it be neural network models [83], equilibrium point models [84], behavioral models [85,86], coupled oscillator models [51,87], kinematic models [88] or models exploiting minimization principles [89,90]. Such an approach would make it possible to enrich or revise certain models on the basis of clinical data, and verify their applicability to the more specific context of ADHD. In turn, this would provide clinicians with information on the relevance of targeting a particular variable, or making predictions about the probability of success of a given therapeutic approach based on theoretical considerations. 

A summary of the aforementioned recommendations for future studies in this field proposed based on the findings of this systematic review are given in Table 5.

## 5. Conclusions

Although handwriting abnormalities in children with ADHD are frequently cited, a systematic review aimed at identifying and collating strong findings of impaired handwriting processes and/or products in this population has been not available to date. Of the 21 articles retrieved, 17 described the quality/legibility of the handwriting of children with ADHD, 12 focused on speed and 14 articles analyzed the process of handwriting. The results reveal that 100% of the studies reported an impaired quality of the written trace and the handwriting process in ADHD individuals, while 25% reported an altered speed of production. The legibility of the produced trace was also found to be the most common type of impairment, whereas the speed of production seemed to be relatively preserved. The prevalence of handwriting deficits in ADHD was not possible to determine on the basis of the studies included. The most general conclusion from our review is that considerable gaps exist in our knowledge of handwriting skills in children with ADHD. Great caution must be exercised when drawing conclusions and more research is needed before making clear statements on whether dysgraphia is actually associated with all children with ADHD. We identified a number of challenges while conducting studies in this field. Most significantly, a wide diversity existed between the experimental conditions or dysgraphia criteria diagnosis, or when verifying other comorbid conditions, ADHD subtypes and medical status (treated or non-treated) of the included children. This evidently calls for standards while conducting studies on the prevalence of dysgraphia in ADHD to ensure case ascertainment, exact co-occurrence rates and comparisons between countries and over time. A summary of recommendations for future studies has been proposed, which might produce reduced heterogeneity and better-quality studies on this issue. It has to be noted that, for enabling comparisons between studies, our review was limited to studies exploring handwriting performance which compared ADHD samples to typically developing children (control groups). In reality, this approach may have excluded some studies investigating the impact of ADHD treatment on dysgraphia, and this important issue also absolutely needs to be addressed. 

## Figures and Tables

**Figure 1 children-11-00031-f001:**
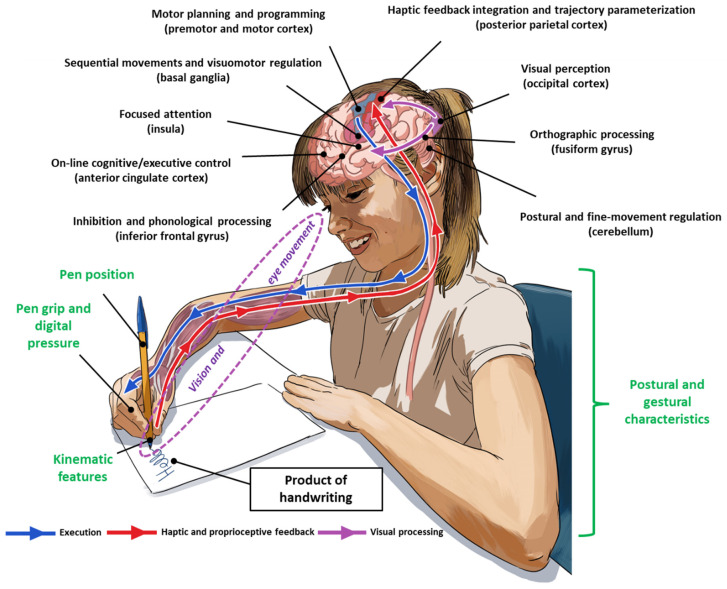
Handwriting is a complex skill involving activations in the left dorsal premotor cortex, the inferior parietal cortex, the fusiform gyrus, the bilateral inferior frontal gyrus, the right cerebellum and the primary motor cortex, which is devoted to manual motor output. In the basal ganglia, the striatum mediates visual–motor integration [18]. Children, unlike adults, recruit in addition the prefrontal cortex, notably the anterior cingular cortex, to perform writing tasks, which is interpreted as a mark of a lower-level automation between the ages of 8 and 11 [19,20]. Handwriting also involves gestural and kinematic characteristics (i.e., the handwriting process in green in the text) leading to the production of the written trace (i.e., the product of handwriting).

**Figure 2 children-11-00031-f002:**
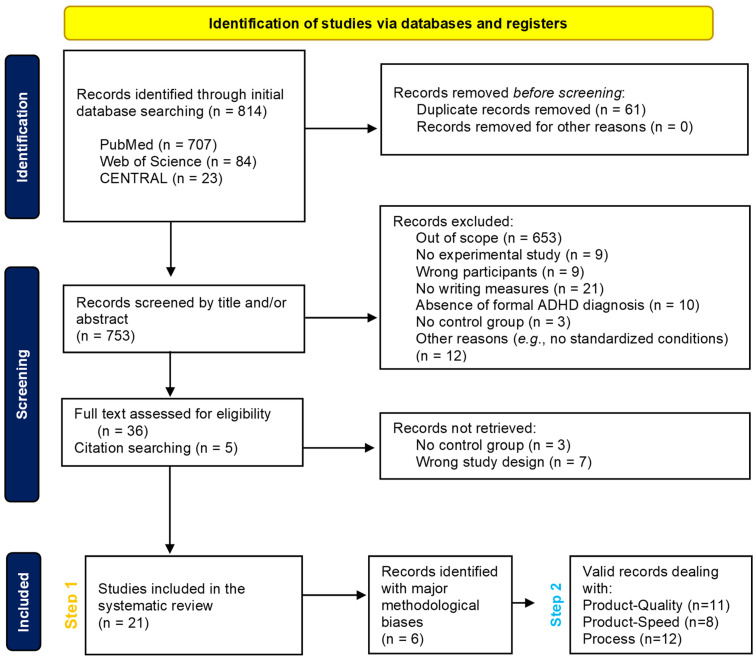
Flow chart of included and excluded studies.

**Figure 3 children-11-00031-f003:**
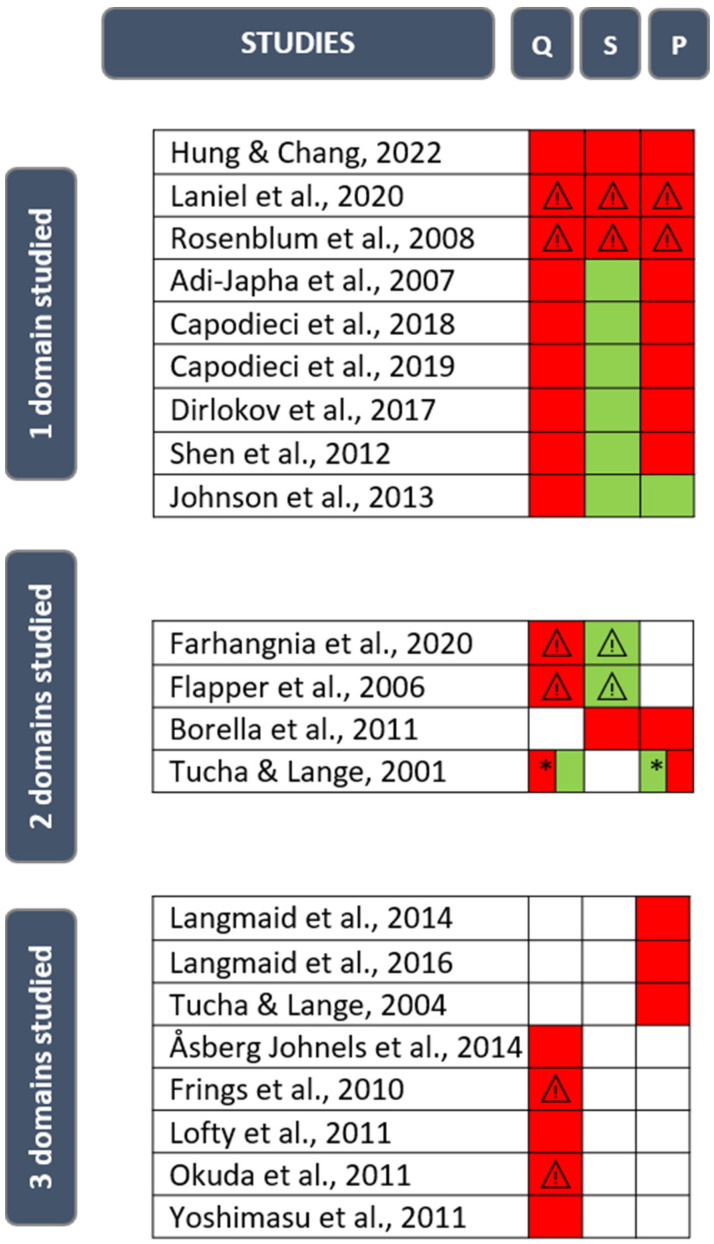
Rapid overview of results of included studies. Abbreviations: Q: quality (product); S: speed (product); P: process; green: non-impaired; red: impaired; ⚠:methodological biases identified; *: without MPH [27,28,30,31,32,35,36,37,38,39,40,41,42,43,44,45,46,47,48,49,50].

**Figure 4 children-11-00031-f004:**
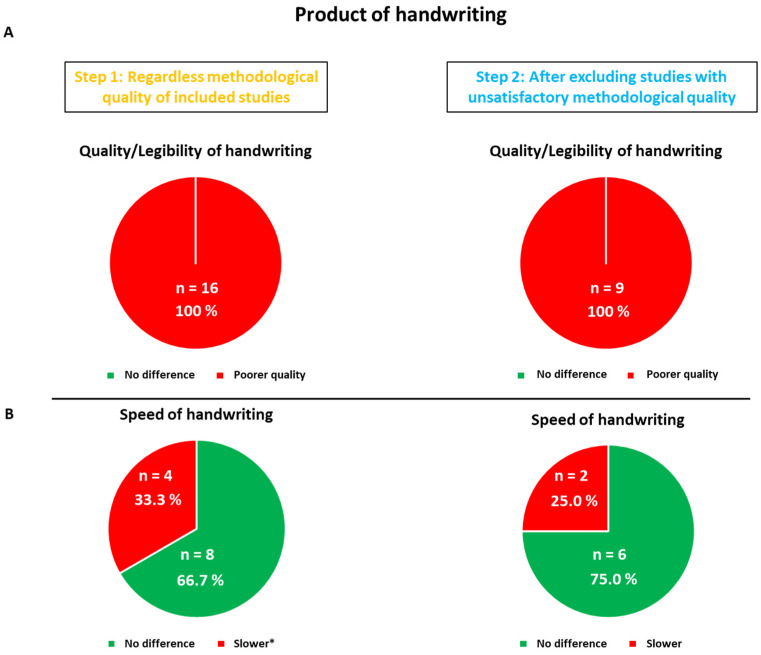
Pie charts for both Step 1 and Step 2 analyses of the product of handwriting. The proportion of studies showing differences between ADHD and typically developing children regarding quality/legibility (Panel (**A**)) and speed (Panel (**B**)) of handwriting before (Step 1) and after (Step 2) the exclusion of unsatisfactory studies due to major methodological biases. *: except for Rosenblum et al.’s study [32], which showed faster production in the ADHD sample.

**Figure 5 children-11-00031-f005:**
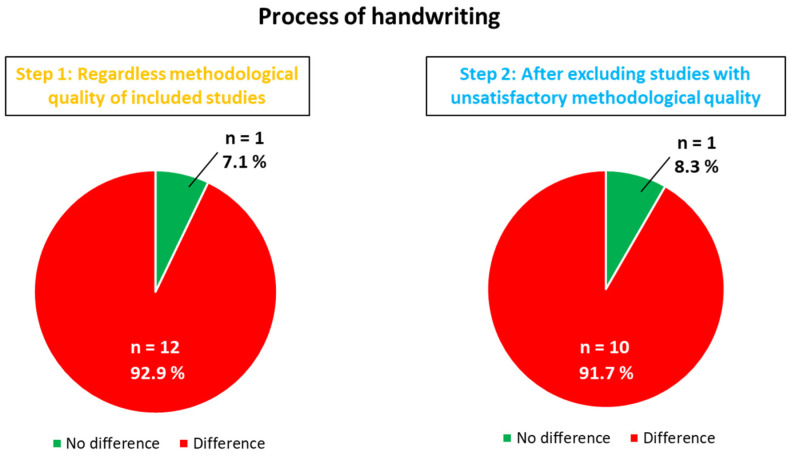
Pie charts for both Step 1 and Step 2 analyses of the process of handwriting. The proportion of studies showing differences between ADHD and typically developing children regarding the process of handwriting before (Step 1) and after (Step 2) exclusion of unsatisfactory studies due to major methodological biases.

**Table 1 children-11-00031-t001:** Critical Appraisal Skills Program (CASP).

Section	Question	Formulation
A: Are the results of the trial valid?	1	Did the study address a clearly focused issue?
	2	Did the authors use an appropriate method to answer their question?
	3a	Were the cases recruited in an acceptable way?
	3b	Was there a sufficient number of cases selected?
	4	Were the control groups selected in an acceptable way?
	5	Was the exposure clearly defined and accurately measured?
	6	Have the authors taken account of the potential confounding factors in the design and/or in their analysis?
B: What are the results?	7	Was the group effect large?
	8	Was the estimate of the group effect precise?
	9	Do you believe the results?
C: Would the results help locally?	10	Can the results be applied to the local population?
	11	Do the results of this study fit with other available evidence?

**Table 2 children-11-00031-t002:** Methodological quality of included studies scored with CASP list for systematic review.

	Methodological Quality							Presentation of Results			External Validity	
	1	2	3a	3b	4	5	6	7	8	9	10	11
Adi-Japha et al., 2007 [28]	Y	Y	Y	Y	Y	Y	Y	C	N	Y	C	Y
Åsberg Johnels et al., 2014 [35]	Y	Y	Y	Y	Y	Y	Y	C	N	Y	Y	Y
Borella et al., 2011 [36]	Y	Y	C	Y	Y	Y	Y	C	C	Y	Y	Y
Capodieci et al., 2018 [37]	Y	Y	Y	Y	Y	Y	Y	Y	Y	Y	Y	Y
Capodieci et al., 2019 [38]	Y	Y	Y	Y	Y	Y	Y	Y	Y	Y	Y	Y
Dirlikov et al., 2017 [39]	Y	Y	Y	Y	Y	Y	Y	C	N	Y	Y	Y
Farhangnia et al., 2020 [40]	Y	Y	C	Y	Y	Y	N	C	N	C	C	Y
Flapper et al., 2006 [41]	Y	Y	Y	N	Y	Y	Y	C	N	C	Y	Y
Frings et al., 2010 [42]	Y	Y	C	N	Y	Y	N	C	N	C	C	Y
Hung and Chang, 2022 [30]	Y	Y	C	Y	Y	Y	C	Y	Y	Y	Y	Y
Johnson et al., 2013 [43]	Y	Y	Y	N	Y	Y	Y	C	N	Y	Y	Y
Langmaid et al., 2014 [44]	Y	Y	Y	N	Y	Y	Y	N	Y	Y	Y	Y
Langmaid et al., 2016 [45]	Y	Y	Y	N	Y	Y	Y	Y	Y	Y	Y	Y
Laniel et al., 2020 [46]	Y	Y	N	N	Y	Y	N	Y	Y	C	C	Y
Lofty et al., 2011 [47]	Y	Y	C	Y	C	Y	C	C	N	Y	C	Y
Okuda et al., 2011 [48]	Y	Y	Y	N	Y	Y	C	C	N	C	C	Y
Rosenblum et al., 2008 [32]	Y	Y	Y	N	Y	Y	C	C	N	C	C	Y
Shen et al., 2012 [27]	Y	Y	Y	Y	Y	Y	Y	Y	Y	Y	Y	Y
Tucha and Lange, 2001 [49]	Y	Y	Y	Y	Y	Y	Y	Y	Y	Y	Y	Y
Tucha and Lange, 2004 [31]	Y	Y	Y	N	Y	Y	Y	C	N	Y	C	Y
Yoshimasu et al., 2011 [50]	Y	Y	Y	Y	Y	Y	N	C	N	Y	C	Y

Abbreviations: Y: Yes; N: No; C: Cannot Tell.

**Table 3 children-11-00031-t003:** Characteristics and results of included studies.

Study	Participants (ADHD and Controls)	Experimental Group	Mean Age (SD)	Gender (Male, Female)	Control Group	Mean Age (SD)	Gender	Inclusion and Exclusion Criteria for ADHD	ADHD Presentation	Medication (Psycho-stimulant)	Handwriting Measures	Q (Quality) S (Speed) P (Process)
Adi-Japha et al., 2007 [28]	40	20	12.2 (5.7)	20 M	20	12.8 (3.6)	20 M	Inclusion criteria: formal diagnosis of ADHD, IQ score > 85, performance within 1.6 SD on a reading test; Exclusion criteria: reading problems.	ADHD/C	Off-state for at least a week before the experiment.	Graphic production on a digitizing tablet; Letters production.	Q and P
Åsberg Johnels et al., 2014 [35]	55	20	10 to 16	20 F	35	10 to 16	35 F	Inclusion criteria: formal diagnosis of ADHD, IQ score > 69; Exclusion criteria: ASD, neuropsychiatric and neurodevelopmental disorders, learning disabilities.	n.s	No medication	FTF; Parental ratings.	Q
Borella et al., 2011 [36]	30	15	9.3 (1.4)	12 M; 4 F	15	9.4 (1.4)	12 M; 3 F	Inclusion criteria: formal diagnosis of ADHD; Exclusion criteria: IQ score < 85, MPH medication, learning disability, mathematical or reading disorders, neurological, sensory, motor, psychiatric or mood disorders.	5 ADHD/I; 10 ADHD/C	No medication	Batteria per la valutazione delle competenze ortografiche nella scuola dell’obbligo; Continuous letters production.	S and P
Capodieci et al., 2018 [37]	32	16	10.5 (6.9)	12 M; 4 F	16	10.1 (6.4)	12 M; 4 F	Inclusion criteria: formal diagnosis of ADHD for only one child, all others on the basis of an ad-hoc questionnaire; Exclusion criteria: Neurological, psychiatric or serious psychological problems; No child had a learning disability.	n.s	No medication	BVSCO-2; Words production.	Q, S and P
Capodieci et al., 2019 [38]	52	26	9.6 (1.2)	22 M; 4 F	26	9.3 (1.1)	n.s	Inclusion criteria: formal diagnosis of ADHD for all but 3 children; Exclusion criteria: neurological or psychological problems, learning disorders.	10 ADHD/I; 10 ADHD/C; 6 ADHD/HI	No medication	BVSCO-2; Dictation tasks; Sentences and words production; Handwriting legibility scale.	Q, S and P
Dirlikov et al., 2017 [39]	167	45	9.9 (1.2)	39 M; 6 F	65	9.9 (1.1)	56 M; 9 F	Inclusion criteria: formal diagnosis of ADHD; Exclusion criteria: intellectual disability, seizures, neurological, chronic medical, genetic, psychiatric (except ODD), speech-related, autistic or psychotic disorders.	7 ADHD/I; 38 ADHD/C	Off-state for at least 24 h before the experiment.	MHA; Copy task.	Q, S and P
Farhangnia et al., 2020 [40]	48	24	8.0 (0.7)	17 M; 7 F	24	8.1 (0.6)	17 M; 7 F	n.s	n.s	n.s	PHAT; Copy task and dictation task.	Q and S
Flapper et al., 2006 [41]	24	12	9.8 (1.7)	11 M; 1 F	12	9.7 (1.2)	11 M; 1 F	Inclusion criteria: formal diagnosis of ADHD+DCD; Exclusion criteria: learning, neurological or psychiatric disorders, IQ score < 70.	6 ADHD/I; 4 ADHD/C; 2 ADHD/HI	Off-state for the first assessment; On-state for 4 to 5 weeks for the second assessment.	BHK; Copy task.	Q and S
Frings et al., 2010 [42]	21	10	12.3 (1.3)	10 M	11	12.1 (1.8)	9 M; 2 F	Inclusion criteria: formal diagnosis of ADHD.	10 ADHD/C	On-state	Copy task	Q
Hung and Chang, 2022 [30]	60	30	7.1 (0.5)	16 M; 14 F	30	7.2 (0.5)	16 M; 14 F	Inclusion criteria: formal diagnosis of ADHD; Exclusion criteria: ASD, seizure disorder, IQ < 80, mental retardation, mood disorders, anxiety or psychotic disorders.	9 ADHD/I; 18 ADHD/C; 3 ADHD/HI	n.s	BCBL; Copy task and dictation task.	Q, S and P
Johnson et al., 2013 [43]	35	14	11.0 (1.95)	14 M	21	11.0 (2.1)	21 M	Inclusion criteria: formal diagnosis of ADHD; Exclusion criteria: medical, sensory, genetic or neurodevelopmental disorders, intellectual disability.	14 ADHD/C	Off-state for at least 24 to 72 h before the experiment.	HPT; Copy task.	Q, S and P
Langmaid et al., 2014 [44]	28	14	10.9 (2.0)	14 M	14	10.6 (2.3)	14 M	Inclusion criteria: formal diagnosis of ADHD; Exclusion criteria: medical, sensory, genetic or neurodevelopmental disorders, intellectual disability.	14 ADHD/C	Off-state for at least 24 to 72 h before the experiment.	Cursive letters production on a digitizing tablet.	P
Langmaid et al., 2016 [45]	28	14	10.8 (2.0)	14 M	14	10.5 (2.2)	14 M	Inclusion criteria: formal diagnosis of ADHD; Exclusion criteria: medical, sensory, genetic or neurodevelopmental disorders, intellectual disability.	14 ADHD/C	Off-state for at least 24 to 72 h before the experiment.	Cursive letters production at 10 mm and 40 mm on a digitizing tablet.	P
Laniel et al., 2020 [46]	25	12	9.5 (1.1)	8 M; 4 F	12	9.9 (1.3)	6 M; 6 F	Inclusion criteria: formal diagnosis of ADHD; Exclusion criteria: Intellectual disability; One child had ODD and an anxiety disorder, another had dyspraxia.	n.s	On-state	BHK; Copy task; Pen-stroke test on a digitizing tablet.	Q, S and P
Lofty et al., 2011 [47]	40	20	7.8 (1.2)	n.s	20	7.8 (1.2)	20 M	Inclusion criteria: formal diagnosis of ADHD; Exclusion criteria: sensory or psychiatric disorders; 60% of included children met criteria for dyslexia.	n.s	n.s	DDS; Copy task.	Q
Okuda et al., 2011 [48]	22	11	8.6 to 11.6	11 M	11	n.s	n.s	Inclusion criteria: formal diagnosis of ADHD; Exclusion criteria: sensory or psychiatric disorders.	n.s	On-state	Scale of dysgraphia	Q
Rosenblum et al., 2008 [32]	24	12	8 to 10	10 M; 2 F	12	8 to 10	10 M; 2 F	Inclusion criteria: formal diagnosis of ADHD; Exclusion criteria: another medical diagnosis.	n.s	On-state	HHE on a digitizing tablet; Copy task.	Q, S and P
Shen et al., 2012 [27]	42	21	8.5 (1.2)	17 M; 4 F	21	8.5 (1.0)	17 M; 4 F	Inclusion criteria: formal diagnosis of ADHD, with possible ODD and CD in the ADHD group; Exclusion criteria: epilepsy, severe anxiety, psychotic disorder, DCD (score < 15th percentile on the M-ABC 2).	6 ADHD/I; 10 ADHD/C; 5 ADHD/HI	Off-state for at least 24 h before the experiment.	THPS; BRWT; Copy task and dictation task on a digitizing tablet.	Q, S and P
Tucha and Lange, 2001 [49]	42	21	10.7 (0.4)	21 M	21	10.5 (0.4)	21 M	Inclusion criteria: formal diagnosis of ADHD, with medication; Exclusion criteria: Concurrent psychotropic medications, ADHD/I or ADHD/HI, reading disability or spelling disorder; Four children had a mathematics disability.	21 ADHD/C	Off-state for at least 10 h before the experiment and on-state 1 h after the administration.	Copy task and dictation task on a digitizing tablet.	Q and P
Tucha and Lange, 2004 [31]	20	10	9.9 (n.s)	5 M; 5 F	10	9.9 (n.s)	5 M; 5 F	Inclusion criteria: formal diagnosis of ADHD, with possible ODD and CD in the ADHD group; Exclusion criteria: neurological and psychiatric disorders.	n.s	On-state	Sentences production on a digitizing tablet.	P
Yoshimasu et al., 2011 [50]	5699	379	10.4 (4.6)	284 M; 95 F	5320	n.s	2666 M; 2654 F	Inclusion criteria: retrospective cohort-based study which has sought formal diagnosis of ADHD (based on DSM criteria); Exclusion criteria: IQ score < 50, written language disorder with or without reading disability.	n.s	Possible medication	Information retrieved from individualized education program goals for written language and/or specific writing subtest scores ≤ 90; Legibility and/or writing subtest scores.	Q

Abbreviations: ADHD: attention deficit hyperactivity disorder; ADHD/C: combined presentation; ADHD/I: inattentive presentation; ADHD/HI: hyperactive–impulsive presentation; ASD: autism spectrum disorders; BCBL: Battery of Chinese Basic Literacy; BHK: Concise Assessment Scale for Children’s Handwriting; BRWT: Basic Reading and Writing Comprehensive Test; BVSCO-2: Batteria di Valutazione della Scrittura e della Competenza Ortografica 2; CD: conduct disorder; DCD: developmental coordination disorder; DDS: Dysgraphia Disability Scale; DSM: Diagnostic and Statistical Manual; F: female; FTF: Five to Fifteen Questionnaire; HHE: Hebrew Handwriting Evaluation; HPT: Handwriting Performance Test; IIV: intra-individual variability; IQ: intelligence quotient; M: male; MHA: Minnesota Handwriting Assessment; MPH: methylphenidate; n.s: not specified; ODD: oppositional defiant disorder; PHAT: Persian Handwriting Assessment Tool; THPS: Tseng Handwriting Problem Checklist; WM: working memory.

**Table 4 children-11-00031-t004:** Main results of included studies.

Study	Main Results
Adi-Japha et al., 2007 [28]	ADHD children made more errors regarding letter insertions, substitutions, transpositions and omissions, produced confusedly shaped letters and replaced the end-of-word letter with its simpler and more common middle-of-the-word version; Speed of handwriting did not differ between groups; No difference considering spatial features; ADHD children displayed poor time utilization, and produced inconsistent and disproportionate writing accompanied by high levels of pressure and multiple corrections; Handwriting problems were associated with attentional problems and reflected an impairment of the graphemic buffer and of kinematic motor production.
Åsberg Johnels et al., 2014 [35]	ADHD children obtained lower scores on parental ratings of handwriting.
Borella et al., 2011 [36]	ADHD children produced fewer writing sequences than the control groups; ADHD children showed greater IIV than control groups.
Capodieci et al., 2018 [37]	No difference between groups considering handwriting speed; In conditions without WM interference, ADHD children produced about 10% fewer graphemes than control groups; In the spatial condition, the difference between groups was slightly greater (−20%), though not statistically significant; In the verbal condition, ADHD children wrote significantly more slowly (−38%) than control groups; The handwriting of ADHD children was generally less legible than the control groups, especially in the verbal and spatial conditions; ADHD group had a higher IIV in the verbal condition than the control groups; High IIV influenced the reduced speed in the case of the verbal condition for both groups.
Capodieci et al., 2019 [38]	ADHD children made more spelling mistakes than control groups in all conditions; ADHD children who better coped with a concurrent verbal WM load had better spelling performance; ADHD children obtained lower scores for handwriting quality than control groups; No difference between groups in terms of writing speed.
Dirlikov et al., 2017 [39]	ADHD children showed worse letter-form scores compared to control groups across conditions (copy, trace and fast trace); No difference in letter-spacing errors between groups; ADHD children made fewer speed inflections across conditions compared to control groups; Both groups showed a significant correlation between letter form and WM performance in the copy condition only.
Farhangnia et al., 2020 [40]	For the copy task, ADHD children had lower global legibility scores compared to control groups; No significant difference between the two groups in terms of space, alignment, size of letters and slant components of writing, as well as for speed of writing; For the dictation task, ADHD children had lower legibility scores compared to control groups, while there was no difference between the groups in terms of space, alignment and slant components and size of letters.
Flapper et al., 2006 [41]	ADHD+DCD children showed lower scores for quality of handwriting, but there was no difference in speed of handwriting between groups; When on-state, of the 11 children with ADHD+DCD who could be assessed a second time, 6 improved their handwriting quality on the BHK, 4 did not improve and one child deteriorated When assessed off-state, ADHD+DCD children did not improve their handwriting speed.
Frings et al., 2010 [42]	Mean letter height did not differ between groups; Letter height increased during repeated writing of the same sentence in the ADHD group only.
Hung and Chang, 2022 [30]	ADHD children had poorer writing performance than the control groups for both character dictation and character copying; ADHD children wrote less fluently and correctly compared to the control groups; Inattention was the stronger predictor of character dictation in ADHD children; Manual dexterity was significantly correlated with character copying in the ADHD group.
Johnson et al., 2013 [43]	ADHD children made more total handwriting errors than control groups (i.e., correction and formation errors); No difference between groups in average height or width; No difference in the coefficient of variability of phrase height and width; No difference in average word spacing; ADHD children included additional strokes more often than control groups; There was a trend towards significant difference between the groups in terms of speed of handwriting, yet it was not significant; In the ADHD group, more corrections were associated with slower handwriting speed and maturational processes contributed to handwriting performance.
Langmaid et al., 2014 [44]	ADHD children were more variable in terms of stroke length and showed inconsistent stroke height when compared to the control groups; No difference in the other kinematic variables; Symptom severity scores were correlated to variability of stroke height (vertical size); Higher scores on the inattentive and total ADHD subtests were significantly correlated with more variable writing size; Stroke duration was significantly related to hyperactive behavior, such that a more hyperactive child had strokes of shorter duration (faster writing); Ballisticity was significantly associated with hyperactive behavior (more ballistic movement when symptoms of hyperactivity increased).
Langmaid et al., 2016 [45]	Despite both groups being significantly more inaccurate on the 40 mm task compared to the 10 mm, ADHD children were unable to maintain the size of their cursive letter at the 40 mm size, contrary to control groups; Groups were comparable on the 10 mm task; ADHD children had more ballistic movements on both tasks; Only pen pressure was positively correlated with inattention scores during the 10 mm task.
Laniel et al., 2020 [46]	ADHD children showed poorer performance on quality scores (BHK), lower writing speed and higher writing size than control groups; On the Pen-stroke test, ADHD children displayed poorer motor planning and execution and greater variability in motor control than the control groups; In the ADHD group only, motor planning on the handwriting task correlated with speed of handwriting on the BHK (the faster a child wrote, the shorter the motor production delay); ADHD children showed greater amplitude of movement on the Pen-stroke test, which was associated with faster motor speed; No relationship between inattention and hyperactivity symptoms with motor control skills was measured for the Pen-stroke test.
Lofty et al., 2011 [47]	A total of 10% of ADHD children had normal handwriting with no disability, 40% had excellent handwriting with a minimum of disability and 50% of ADHD children showed mild to moderate handwriting disability; ADHD children had poorer performance in respecting lines, spacing between words, letter direction, spelling a sentence and punctuation (item of the DDS); No difference between males and females in the ADHD group only on DDS scores; No correlation between DDS scores and age in the ADHD group only.
Okuda et al., 2011 [48]	ADHD children manifested poorer performance regarding flowing lines, descending lines, retouched letters, curvatures and angles of “m”, “n” and “u” letters; They produced more collisions and adherences, sudden movements, irregular sizes and incorrect forms of letters.
Rosenblum et al., 2008 [32]	Poorer performance of ADHD children on most handwriting process and product measures when off-state versus on-state; When off-medication, ADHD children showed more total time and more in-air time than when on-medication and compared to control groups; No difference in handwriting speed when on-state and off-state were compared, while on-state and off-state ADHD children wrote faster than control groups; No difference in product handwriting between on-state and off-state, but ADHD children, regardless of on- or off-state, differed in comparison to control groups.
Shen et al., 2012 [27]	ADHD children scored lower on THSPC and on BSRWT; Despite the speed of writing per se being no different between the two groups, ADHD children spent more on-paper time on the copy task and, hence, needed more time to finish a copy task.
Tucha and Lange, 2001 [49]	When off-state, the quality of handwriting specimens of hyperactive boys was poorer than on-state but more fluent; When off-state, ADHD children did not differ from control groups in handwriting movements; Hyperactive behavior improvement through MPH was associated with increased legibility and greater accuracy of handwriting.
Tucha and Lange, 2004 [31]	When on-state, ADHD children displayed significantly more inversions in the direction of their velocity profiles than control groups; When off-state, there was no difference between the groups; The medication resulted in increased dysfluency during handwriting.
Yoshimasu et al., 2011 [50]	ADHD girls tended to have a single specific writing difficulty, whereas ADHD boys were more likely to have multiple writing difficulties (e.g., legibility + poor paragraph organization).

Abbreviations: ADHD: Attention Deficit Hyperactivity Disorder; BHK: Concise Assessment Scale for Children's Handwriting; BRWT: Basic Reading and Writing Comprehensive Test; DCD: Developmental Coordination Disorder; DDS: Dysgraphia Disability Scale; IIV: Intra-Individual Variability; MPH Methylphenidate; THPS: Tseng Handwriting Problem Checklist; WM: Working Memory.

**Table 5 children-11-00031-t005:** Recommendations for future research aimed at investigating handwriting skills in ADHD populations.

Recommendations	Level of Importance
Eligible Population	
Ensure a reasonable sample size necessary to conduct the study.	High
Verify the diagnosis of ADHD with a formal diagnosis following DSM-5 indications and the use of gold-standard tools.	High
Verify the diagnosis of HD using standardized, valid and reliable tools.	High
Harmonize the comparator group(s) with previous studies to facilitate comparison:	
ADHD only;	High
HD only;	High
Typical.	High
Harmonize the reference group with previous studies to facilitate comparison:	
ADHD + HD.	High
Ensure children acceptability (motivation to study participation).	Low
Subgroup Analysis	
Consider demographic characteristics:	
Age;	High
Gender;	High
Handless;	Low
IQ.	High
Socioeconomic factors.	Low
Ethnicity.	Low
Cultural background.	Low
Include documentation of ADHD subtypes.	High
Screen for comorbid emotional or behavioral conditions (e.g., anxiety, depression, sleep disturbance).	High
Screen for comorbid neurodevelopmental conditions (e.g., learning and language disorders, autism spectrum disorders).s	High
Screen for comorbid physical conditions (e.g., tics).	Low
Treatment and Care	
Considered ADHD specific treatments.	
Methylphenidate.	High
Other medication.	Low
Behavioral interventions for treatment of ADHD or comorbidities.	High
Motor behavioral interventions (psychomotricity).	High
Expert Panel	
Harmonize measurement of key handwriting elements to facilitate pooling of and comparisons between study findings.	High
Use common outcome measures to facilitate pooling of and comparisons between study findings.	High
Supervise the experimental handwriting testing without knowing the child’s group (blind test).	Low
Assess the handwriting performance without knowing the child’s group (blind evaluation).	High
Assess the handwriting performance by an expert panel of experiment judges (two or more).	High
Theoretical Considerations	
Discuss clinical findings with more fundamental work addressing the theoretical models of handwriting applied to the specific context of ADHD.	High

## Data Availability

Not applicable.

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
