# Peer review of "Is There a Deficit in Product and Process of Handwriting in Children with Attention Deficit Hyperactivity Disorder? A Systematic Review and Recommendations for Future Research"

_children, 2023, doi:10.3390/children11010031_

Round 1

Reviewer 1 Report

Comments and Suggestions for Authors

Very well-written, great figures and easy read.

In the discussion the authors may look into the 4-6 types of ADHD (e.g. https://www.ncbi.nlm.nih.gov/pmc/articles/PMC4131943/), and also how Q, S and P would be affected for AHDH-H vs ADHD-I

which might explain the inconclusive data, i.e. subtype 1 + subtype 2 may "cancel each other out" at the group level

Some minor issues

line 73: cognitive abilities (e.g., working memory, inhibition) posture, <- comma after bracket missing

Table 4: font change, please also write out IIV (as this abbreviation is  established solely in the legend of Table 3)

I assume Batteria per la valuta-zione delle compe-tenze ortografiche nella scuola dell’ob-bligo; Continuous let-ters production is BVSCO-2 ?!?! If so, the written out version in row 3 is not necessary

to make it even easier for the reader, please consider in Table 3 to mark in the last column behind each measure whether it is Q, S or P. Thanks

figure 4: please use instead of difference in the quality a directional information, i.e. no difference vs poorer quality 

same for speed of handwriting: indicate whether it is slower or faster in ADHD, not just "difference"

line 22 "3.1. roduct of Handwriting Results" P missing

line 144 "ADHD children therefore clearly seem to experience problems both in the product and process of handwriting" - maybe be a bit more nuanced as it is (100% and 25% and 92% difference for Q, S and P) or follow-up with it in the next sentence

indeed, the data suggests an aberrant speed-accuracy trade-off in ADHD

line 187: "At best, medication seems effective for a portion of children (e.g., Brossard-Racine et al., 2015) while at worst, there is no impact at all (e.g., Rothe et al., 2023)." <- you previously cite that the process was worse, so worst is not no impact but worse process or? 

Reviewer 2 Report

Comments and Suggestions for Authors

Thank you for the opportunity to review this systematic review of handwriting problems in ADHD.  I appreciate the authors' diagram and deconstruction/operationalization of handwriting into production and process dimensions, which will make for better, more rigorous research in future.  I think they would be better served by framing their paper (and future work) in terms of Handwriting Problems, as "Handwriting Disorders" quickly get confounded with the conditions in which handwriting problems are observed (ADHD, DCD, Specific LD in Written Expression, etc.).  Also, SLD in Written Expression includes an element of mentally organizing one's written work that can also get quickly confounded with inattention and other executive functioning problems.  So keeping the manuscript at the level of problems rather than disorders would be wise.  Because the authors evoke the possible benefits of methylphenidate, I would commend them to read and cite this systematic review: Froehlich T, Fogler J, Barbaresi W, Elsayed N, Evans S, Chan E.  Using ADHD Medications to Treat Children with Co-Existing ADHD and Reading Disorders: A Systematic Review.  Clinical Pharmacology & Therapeutics. 2018; 104(4):619-637. doi: 10.1002/cpt.1192.  Given their initial discussion of brain mechanisms, I was surprised that neuroimaging techniques did not figure more into their discussion.  

Author Response

All comments and suggestions have been taken into account (in red in the text).

We systematically removed the term "disorder" after "handwriting", and replaced it by the terms "disturbances", "deficits", "problems", or "abnormalities". The reference Reviewer #2 suggested (Froehlich et al., 2018) has been added in the Discussion/recommandation section with an associated sentence, and in the list of references too. 

We sincerely thank the Reviewer for his/her help to improve our manuscript. 
